# Building and Architectural Details of Tenement Houses Built at the Turn of the 19th and 20th Centuries in Central European Region—Hygrothermal Analysis

Klara Kroftova *[ID] and Radek Zigler [ID]

Faculty of Civil Engineering, Czech Technical University in Prague, 166 29 Prague, Czech Republic
* Correspondence: klara.kroftova@fsv.cvut.cz

**Abstract:** Restoring historic buildings is a challenging task in an environment where any insensitive or unprofessional intervention can cause irreparable damage. Among the most important demands currently placed on the construction industry are the protection of structural details, materials and technologies, and the extension of the life of these historic buildings. In this context, we should mention the protection of the high number of tenement buildings in European cities from the second half of the 19th and early 20th centuries, whose structural quality is relatively high and where many other building details and elements have been preserved. The brick dwellings of the period, which are between 85 and 170 years old, do not fully comply with many of the requirements and provisions of the current regulations and standards. The serious shortcomings of brick tenement buildings include, among other things, the inadequate thermal resistance of the envelope and infill structures and the high energy consumption of the operation of these buildings. This paper focuses on analysing this situation and defining the requirements for renovation, while preserving the architectural and historical values of urban buildings; achieving acceptable compliance with the requirements and provisions of the currently applicable regulations and standards; and demonstrating cost-effectiveness.

**Keywords:** tenement building; brick dwellings; building physics; thermal resistance; heat transfer coefficient; hygrothermal analysis

## 1. Introduction

The development of architecture throughout the so-called long century (1790–1914) in Central Europe was influenced not only by political events, but also by cultural and social developments that fundamentally changed the understanding of the world. The development of industry and economy, which led to people moving to cities in search of work, had a significant impact on the development of architecture and, especially, urban construction.

Urban tenement houses, which were built in the second half of the 19th and the beginning of the 20th century in the large cities of the Czech Kingdom, form an important part of urban complexes. Many of them, however, have defects and faults that damage their expression and, in many cases, also threaten the useful properties of the building. It can be summarised that defects and failures in the structures of urban tenement buildings are most often caused by design defects; defects in materials; and degradation processes caused mainly by damp, misuse and neglected maintenance (Figure 1). The cause of the deterioration of the required properties of the materials and structures is caused, on the one hand, by the specific properties of the building materials (composition, structure, etc.) and, on the other hand, by the time-varying parameters of the external environment (temperature, humidity, etc.) affecting the buildings and their parts. These parameters, together with the material parameters, create conditions that initiate or accelerate mechanical, mineralogical, physical, chemical and biological degradation processes.

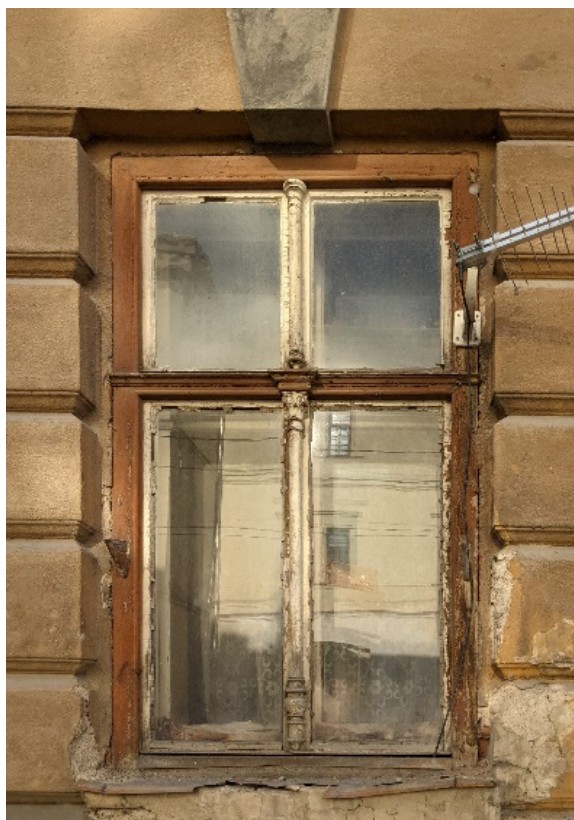
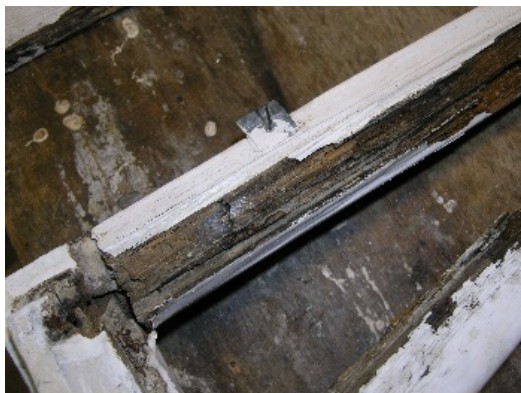
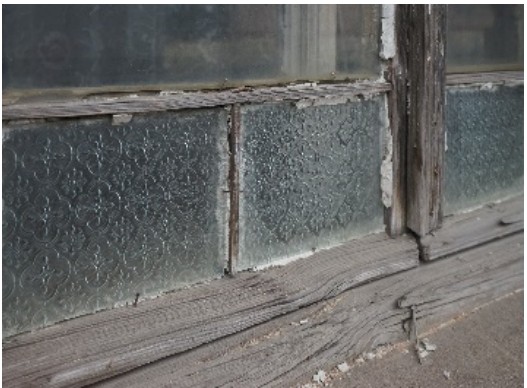

**Figure 1.** Examples of windowpanes in urban tenement buildings with neglected maintenance, mainly showing failures caused by cyclical changes in atmospheric influences (Photographs by K. Kroftova).

The topic of defects and failures of urban tenement houses is very broad; this paper only focuses on issues related to the thermal properties of the envelope, respectively, i.e., horizontal and infill structures. Incorrect solutions of defects and failures of these building components can have a significant negative impact not only on the appearance of buildings, but especially on their internal environment.

According to data from the Czech Statistical Office (2021), approximately 10% of the housing stock consists of buildings built before 1919. The reconstruction of these historic buildings holds considerable potential for meeting the Kyoto Protocol targets [1] and can significantly support the objectives of long-term sustainable development. The renovation of existing apartment buildings (including those built before 1945, i.e., approximately 25% of the housing stock) to an energy efficient standard can reduce building energy demand by up to 70% [2]. Modifying the structural and physical properties of the envelope of historic urban buildings should be part of a Europe-wide sustainability strategy [3].

The conservation of a large number of 19th and early 20th century buildings is a very important topic. Urban residential buildings from this period provide a plastic image of the lifestyle of their time, while at the same time, the principles and technologies of traditional materials, construction and craftsmanship are still preserved in their structural elements. It is advisable to protect all elements of these buildings and to find methods of restoring them efficiently so that they meet current building and technical requirements while not losing their historical values (e.g., mullioned windows, stucco decoration of facades, roof cladding, etc.).

Unfortunately, the question of the approach of improving the quality of thermal insulation properties of building envelopes from the second half of the 19th and early 20th centuries is currently not sufficiently addressed in the Czech Republic. Moreover, this issue is significantly complicated by the nature of the facades of urban residential buildings,

which used a wide range of architectural expressive means: cornices, suprafenesters, stucco decoration and many others. In most cases, existing works deal with a more general theme (e.g., dealing with buildings of general architectural value [4–6]) or are narrowly focused on a specific building or part of a building (e.g., window replacement [7]).

This paper should, in particular, draw the attention of architects and engineers to this complex issue, which must be addressed comprehensively, taking into account all contexts—not only thermo-technical, but also architectural and utilitarian.

The issue of the restoration of these buildings also places demands on the awareness of the owners, as these buildings are often not classified as cultural monuments but are located in a historic, protected environment; therefore, they should be given increased attention with regard to the requirements of heritage protection.

## 2. Materials and Methods

This paper is divided into three sections. The first section is focused on the characteristics of the designed structures of urban tenement houses in the Czech Republic, which is based on the study of historical documents and on a field survey carried out on selected tenement houses in Prague. In the second section of the paper, we refer to analogous examples from foreign studies focused on the hygrothermal properties of residential buildings. The third section presents an analysis of selected details of building structures—specific critical points that are part of the thermal envelope of apartment buildings. The analyses were carried out by means of various numerical models.

### 2.1. Building Structures

With the development of urban construction, the need for regulation has also grown. Throughout the 19th century, some of the most important areas of building law were clarified, especially in relation to fire safety [8]. The building codes determined, for example, the maximum height of buildings and the design of ceiling structures, staircases or roofs, etc.; these were valid until the end of the Austro-Hungarian Monarchy and were largely adopted in the legal system of Czechoslovakia after the establishment of the independent state (1918). However, buildings were still designed on the basis of the builder's experience, with the thickness of the vertical load-bearing structures determined by empirical formulae depending on the number of storeys, the height of the storey and the depth of the wing.

Magazines and building manuals, such as the *Allgemeine Bauzeitung*, published in Vienna, from 1836 [9], or Jöndl's *Instruction on Civil Engineering*, from 1840 [10], also played an important role in the development of building structures. These and other similar publications conveyed information about new materials and technologies and promoted changes in building typology and design.

### 2.1.1. Vertical Structures

Urban residential masonry buildings were characterized by massive walls, most often implemented as brickwork with mixed masonry and especially with solid brickwork. From the second half of the 19th century onwards, on the top floors was required a wall thickness of at least 45 cm, and this increased downwards (Figure 2). At the turn of the 19th century, and especially from the beginning of the 20th century, there was an increase in the number of buildings whose walls were already composed of hollow burnt bricks on the upper floors or, later, of brick cavity blocks comprising lightweight concrete, etc. (cinderblock or slag pumice). Brick walls and pillars of greater thickness, usually over 70 cm, were often constructed as multi-layered. Lime mortar was predominantly used as a bonding material until the middle of the 19th century, followed by an increase in the use of lime-cement mortar in the next period, and then the use of cement mortar in the 20th century [11].

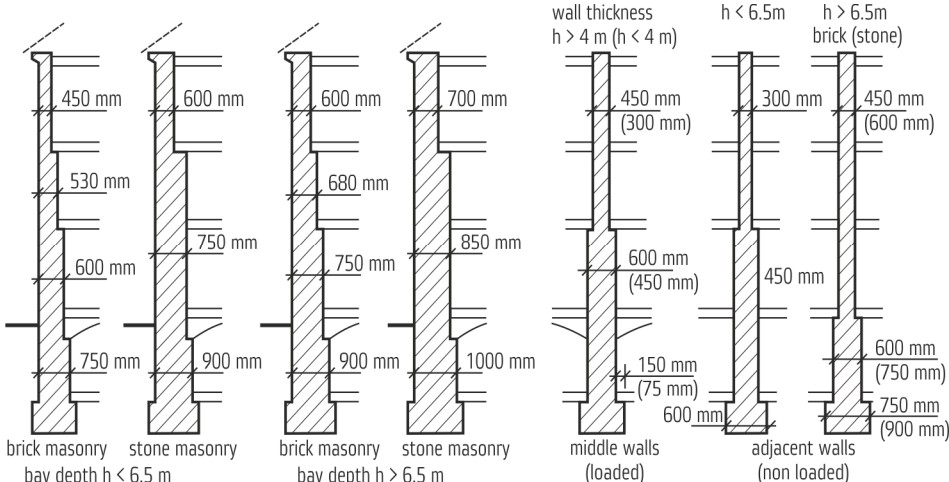

**Figure 2.** Graphical representation of the increasing thickness of the vertical structure in accordance with the requirements of the *Building Code for the Royal Capital City of Prague and its Suburbs of 1886* (Drawn by authors, based on [12]).

> *"The determination of the true thickness of a wall in civil engineering is, in most cases, the result of careful consideration of various factors, which are: 1. the kind of material, 2. the shape of the wall, 3. the thoroughness of the work, 4. the magnitude, direction, and action of the forces acting, 5. the free length and height of the wall, 6. the number and size of openings, 7. the position of the building, 8. the purpose of the wall".* [13]

The stability of the load-bearing system of the multi-storey masonry buildings was ensured by longitudinal walls on which the floor structures were placed, interconnected with the longitudinal walls (Figure 3), [13]. To ensure stability, the transverse walls—gable and stair walls and masonry partitions of 15 cm and 30 cm thickness—played an important role in connecting the longitudinal, primarily load-bearing walls, ensuring their stability and contributing to the spatial interaction of the load-bearing system [14].

Balconies, pavilions and bay windows were also characteristic elements of the town's architecture, and their design (dimensions) had to meet the requirements of the building regulations (Figure 4). In most cases, the structure of balconies and pavilions consisted of arches or slabs (stone, ceramic, or wooden) placed on brackets or beams. The bay windows often dominated the façade or corners of the house. They were constructed with a supporting structure consisting of an iron ('traverse') frame, the beams of which were, for example, extended beams of the ceiling structure, or special beams anchored into the central walls. The 30–45 cm thick perimeter walls of the bay windows were made of perforated and lightened hollow bricks, in some cases supplemented with heraclite or pressed cork boards (to improve thermal insulation properties).

### 2.1.2. Horizontal Structures

In most of the urban tenement buildings in the area, it is possible to mainly encounter various types of wooden beam ceiling structures. Among the horizontal structures used, the dominant ones were spalled and semi-salvaged ceilings (i.e., with a minimum 80 mm thick embankment at the upper face, while the protection of the wooden beam structure at the lower face usually consisted of underfloor boards with reed plaster). Among the structures used, wooden beam ceilings with a backfill were predominant.

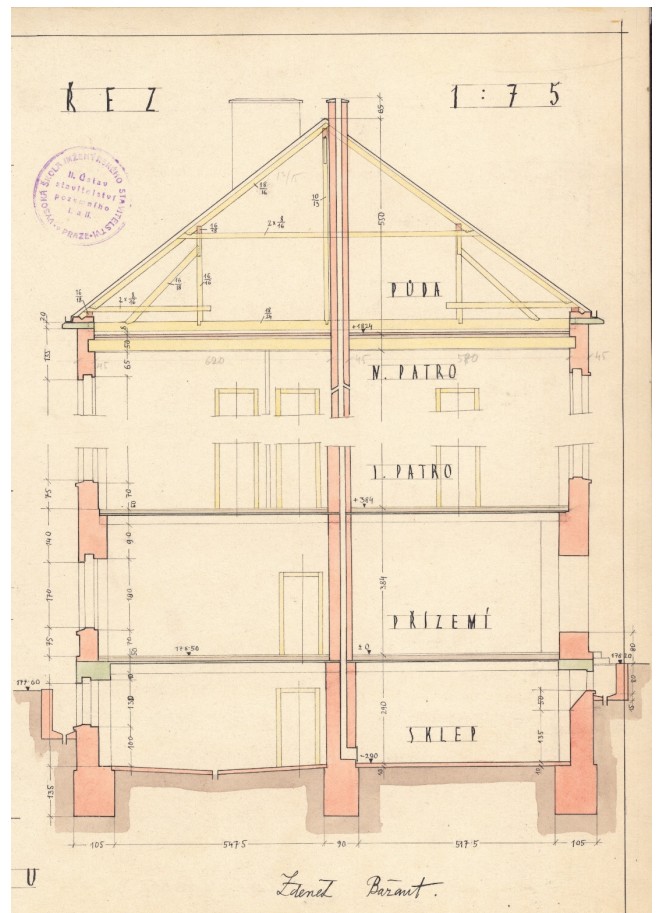
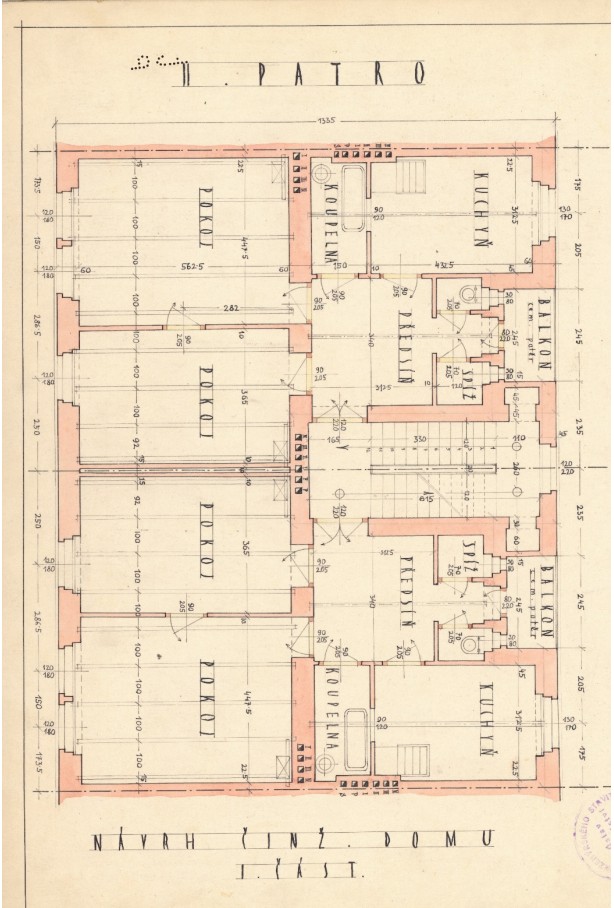

**Figure 3.** Section of the staircase and ground plan of an apartment building, school work Z. Bazant, 1926 (Explanatory notes: ŘEZ = Section, PŮDA = Attic, PATRO = Floor, PŘÍZEMÍ = Ground floor, SKLEP = Cellar, POKOJ = Room, KUCHYŇ = Kitchen, PŘEDSÍŇ = Lobby, BALKON = Balcony, KOUPELNA = Bathroom, NÁVRH ČINŽ. DOMU I. ČÁST = Apartment house design I. part) [15] (public domain).

In areas with high fire safety requirements, e.g., on the ground floor of tenement buildings, mostly fireproof ceilings were designed. Vaults were considered to be safe, fireproof ceiling structures (or were required by the building regulations), which were used in entrances, house entrances, cellars and, if necessary, in staircases, corridors and on the ground floor of tenement buildings.

An important function in terms of ensuring spatial rigidity, stability and resistance to the effects of forced reshaping were the so-called wall and beam clamps. From the beginning of the 19th century, iron tongs were used in walls and vaults to stiffen and retract the building and to absorb the oblique pressures, especially when the masonry was settling and the foundation soil was being pressed.

### 2.1.3. Window Fillings

The most important parts of the tenement houses are the windows, which co-create the architectural expression of the buildings. The material and technical design of the window openings and their fillings represent a historical document, which brings closer the technical, craft and artistic level of the individual professions that participated in its creation at the time—carpenters, glaziers, locksmiths, stonemasons, masons and others [16]. For these reasons, informed restoration should be preferred to replacement with modern infill so as to preserve the expression and integrity not only of a single building, but also of

entire street fronts. In the case of listed buildings, the cultural value in exposed locations may take precedence over the thermal quality of the building.

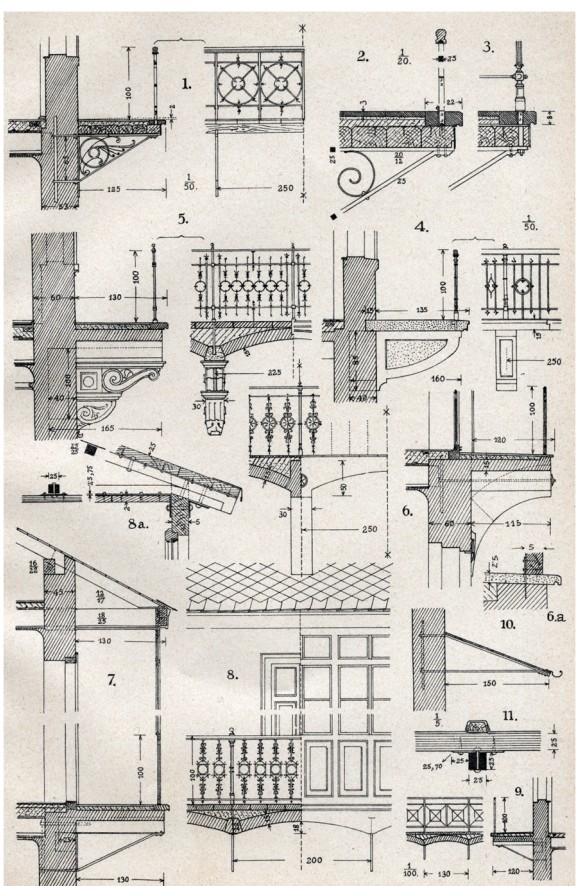
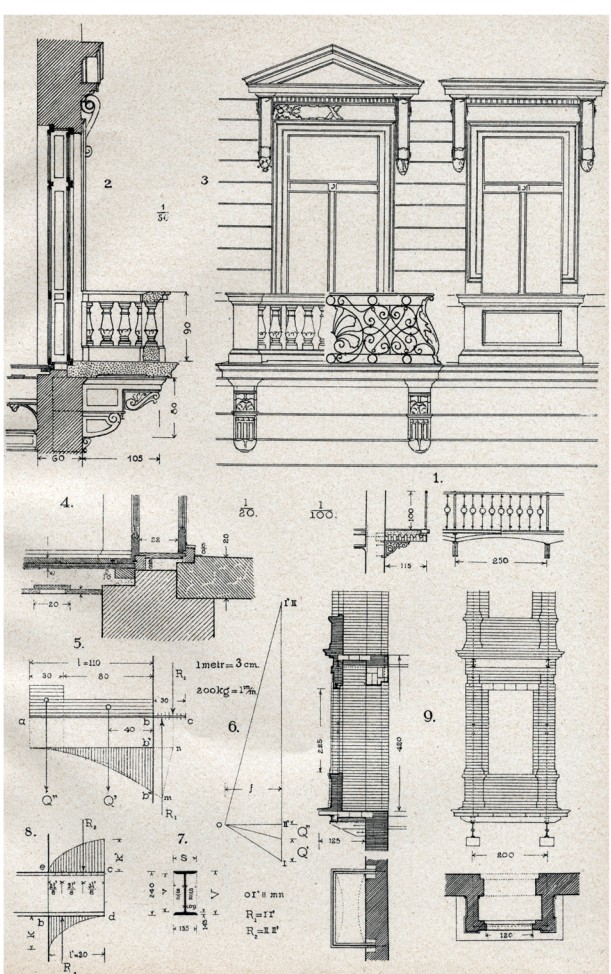

**Figure 4.** Characteristic details of the design of balconies, pavilions and bay windows on a townhouse listed in Pacold's *Construction Manual* from the early 20th century [13] (public domain).

From the typological point of view, several basic types of windows were used in the Austro-Hungarian urban construction of the turn of the 19th and 20th centuries, based on the general design principle of the window [17,18].

The simplest window was a single window frame. In connection with the greater demands on the thermal comfort of interiors, from the beginning of the 19th century onwards, second window frames were first added to existing windows (so-called winter windows), and in new buildings, double glazing was introduced (Figure 5b). From the mid-19th century, the double window was the standard for heated interiors, while the use of the single window was restricted to unheated and service areas (e.g., corridors, staircases, lavatories, pantries, workshops, etc.).

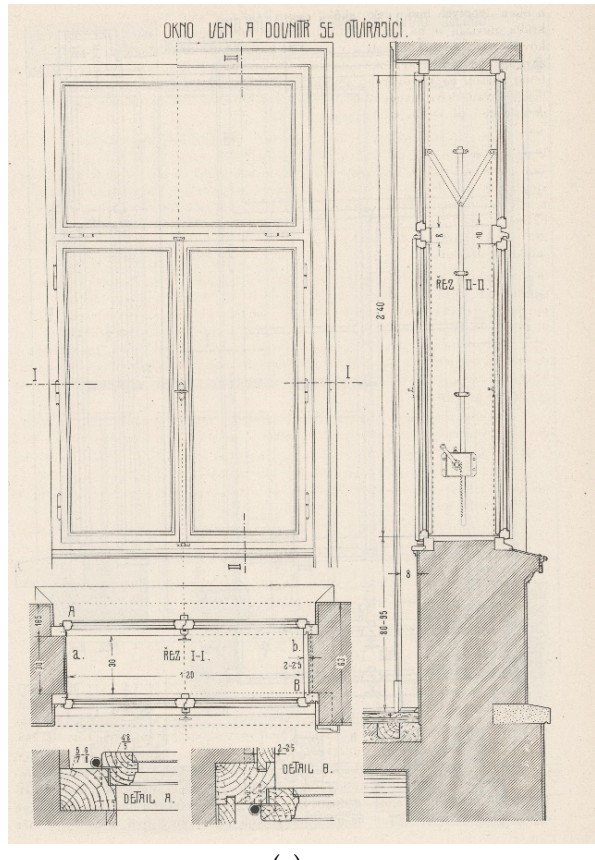
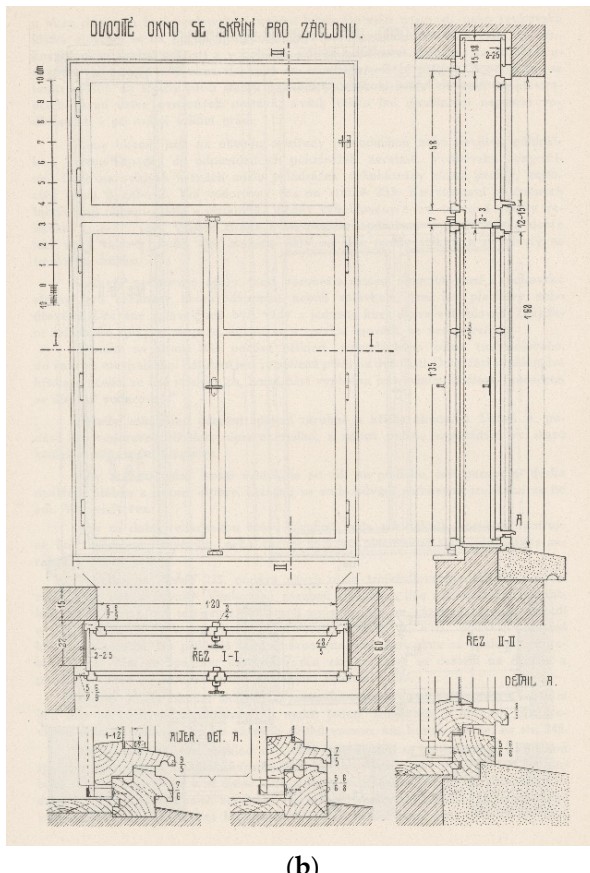

(**a**)  (**b**)

**Figure 5.** Details of fenestration structure [18]: (**a**) Double-hung window with outward-opening sashes of outer frame; upper and lower sashes in younger design without divided glass panes. (**b**) Double-hung window with inward-opening sashes recessed into depth of jamb (Explanatory notes: OKNO VEN A DOVNITŘ SE OTVÍRAJÍCÍ = Window opening out and in, DVOJITÉ OKNO SE SKŘÍNÍ PRO ZÁCLONU = Double window with cabinet for the curtain, ŘEZ = Section), (public domain).

The windows consisted of two separate frames—an outer and an inner one (the sashes of the inner window opened inwards and the sashes of the outer frame, facing the facade of the building, opened outwards) and had a number of drawbacks, including the exposure of the outer sashes to the weather and their rapid degradation. In order to protect the outer frame, it was recessed into the window opening and, at the same time, the inward opening of the sashes was reversed. Thus, in the last quarter of the 19th century, a double-paned window was created by joining the outer and inner windows into a single structural unit using wooden boards (jambs) (Figure 5a). This step, among other things, made it easier to fit into the building opening without the need to laboriously measure the correct position of the window frames in relation to each other.

> *"The size of the windows is determined by the dimensions of the room to be illuminated by them, mainly by their depth and height, then by the desired intensity of lighting and finally by the overall architectural conditions on the exterior of the building. I will give 1/4 of the floor area, what the light area of the windows, sharp, 1/6 medium, 1/8 dim lighting".* [18]

### 2.2. Hygrothermal Properties of Residential Buildings

The serious shortcomings of the brick tenement houses from the second half of the 19th and early 20th centuries include the unsatisfactory thermal resistance of the envelope structures and the high energy consumption of the operation of these buildings, which is several times higher compared to the current structures. The thermal assessment of

the envelope brick walls of the historic buildings demonstrated their insufficient and unsatisfactory thermal properties in terms of the currently applicable standard CSN 73 0540-2: 2011 [19].

Since 1949, when CSN 1450 [20] came into force and defined the value of the thermal resistance of the envelope structure, the requirement for the value of the thermal resistance of the envelope structure (according to the current CSN 73 0540: 2011) has increased by at least 4 times (not considering the requirement for passive standard from 2020). Figures 6 and 7 show a comparison of the thermal performance of the envelope masonry wall with the evolution of the standard's requirements. The first standard that formulated requirements for the thermal insulation properties of building structures in the Czech Republic, CSN 73 0540 "Thermal properties of building structures and houses", was published in 1964. The minimum value of the thermal resistance of the perimeter wall in this standard was based on the provisions of the Building Code for the Capital City of Prague from 1886, which required that perimeter wall structures should be at least 450 mm thick [21]. This thickness corresponds approximately to a thermal resistance of $R_N = 0.6$ m$^2$K/W, which is a satisfactory value as required by that standard.

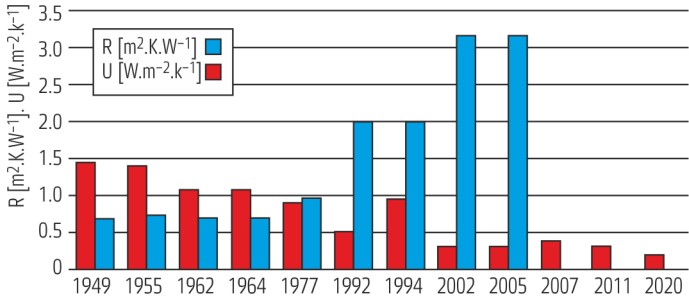

**Figure 6.** Development of the standard's requirements for thermal resistance and heat transfer coefficient of vertical envelope structures from 1949 to 2020. (Analysis by the authors).

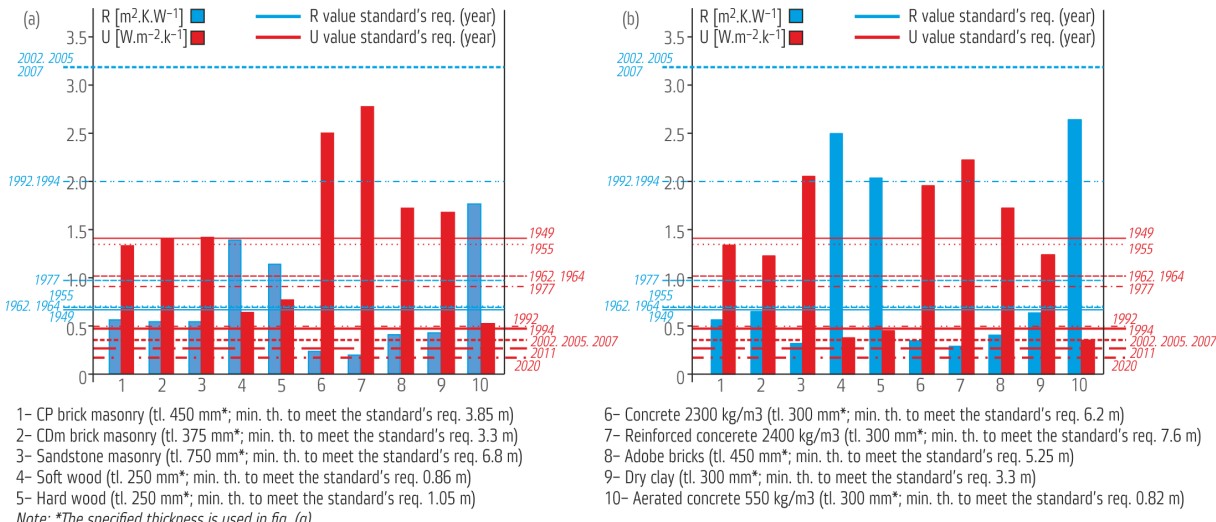

1– CP brick masonry (tl. 450 mm*; min. th. to meet the standard's req. 3.85 m)
2– CDm brick masonry (tl. 375 mm*; min. th. to meet the standard's req. 3.3 m)
3– Sandstone masonry (tl. 750 mm*; min. th. to meet the standard's req. 6.8 m)
4– Soft wood (tl. 250 mm*; min. th. to meet the standard's req. 0.86 m)
5– Hard wood (tl. 250 mm*; min. th. to meet the standard's req. 1.05 m)
*Note: \*The specified thickness is used in fig. (a)*

6– Concrete 2300 kg/m3 (tl. 300 mm*; min. th. to meet the standard's req. 6.2 m)
7– Reinforced concerete 2400 kg/m3 (tl. 300 mm*; min. th. to meet the standard's req. 7.6 m)
8– Adobe bricks (tl. 450 mm*; min. th. to meet the standard's req. 5.25 m)
9– Dry clay (tl. 300 mm*; min. th. to meet the standard's req. 3.3 m)
10– Aerated concrete 550 kg/m3 (tl. 300 mm*; min. th. to meet the standard's req. 0.82 m)

**Figure 7.** (**a**) Comparison of thermal properties (thermal resistance, heat transfer coefficient) for selected types of vertical envelope structures with the standard's requirement; (**b**) comparison of thermal properties (thermal resistance, heat transfer coefficient) of vertical envelope structures 450 mm thick composed of selected materials with the standard's requirement. (Analysis by the authors).

Thermal standards responded to the global energy crisis in the 1970s by increasing the requirements for the thermal insulation capacity of building structures, especially envelope structures. This development began with the revision of the thermal engineering standard

in 1977 and has essentially continued to the present day. Standard CSN 73 0540-2, "Thermal protection of buildings. Part 2: Requirements", of 2011, specifies for the external wall the required value of the heat transfer coefficient at the level of $U_N = 0.30$ W/m²K for both light and heavy structures. The recommended values for heat transfer coefficients are 0.20 and 0.25 W/m²K, respectively (passive houses U = 0.18 to 0.12 W/m²K). It is, therefore, clear that there has been an increase of approximately fourfold (for the required values) to ninefold (for the recommended values or the values for the passive standard) compared to the values of the 1960s. The current thermal standard CSN 73 0540: 2011 specifies three values for the heat transfer coefficient of individual building structures: the required value, the recommended value and the range of values recommended for passive buildings. The required value of the standard heat transfer coefficient of the structure is a value that must not be exceeded and which guarantees the function of the structure without any structural and physical defects. Decree No. 78/2013 Coll. (replaced in 2020 by Decree 264/2020 Coll.) on the energy performance of buildings from 2013 requires that the resulting heat transfer coefficient of modified structures is lower than the recommended value, according to CSN 73 0540-2: 2011. The majority of the existing brick urban tenement buildings do not meet the required values of heat transfer coefficient and energy performance (Figure 8).

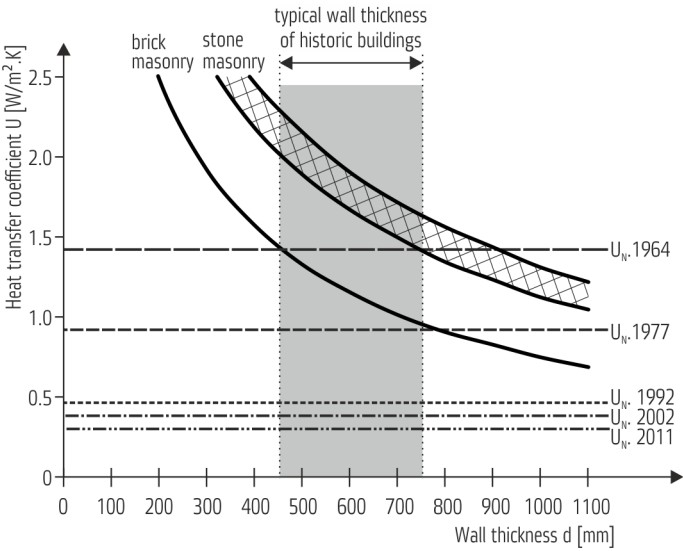

**Figure 8.** Requirements for heat transfer coefficient, according to CSN 73 0540 and CSN 73 0540-2 (brick masonry ρ = 1800 kg/m³, λ = 0.86 W/m. K, stone masonry ρ = 2400–2800 kg/m³, λ = 1.4–1.7 W/m. K). (Analysis by the authors).

The Energy Performance of Buildings Directive [22], together with the Energy Efficiency Directive [23], promote policies for achieving efficient and decarbonised building stock (the goal is by 2050), creating a stable environment for investment decisions and enabling consumers and businesses to make more informed choices in order to save energy and money. The last proposed revision reflects even higher ambitions in terms of climate and social actions and sets a goal for achieving a zero-emission and fully decarbonized building stock. Among the main steps to achieve this goal, there is the gradual introduction of cost-optimal minimum energy performance standards (even for existing buildings); a definition of deep renovation and the introduction of building renovation passports; the modernization of buildings and their systems; better energy system integration (for heating, cooling, ventilation); and others. The Renovation Wave Strategy [24] aims to at least double renovation rates in the next ten years and ensure that renovations lead to higher energy and resource efficiency. This should enhance the quality of life for people living and using these buildings, reduce energy poverty, reduce Europe's greenhouse gas emissions, develop digitalization and improve the reuse and recycling of materials. All these directives and actions are focused on governmental and public building, but their implementation

in the private sector (i.e., residential buildings) is encouraged. Moreover, the operative energy consumption should not be the only aspect of the evaluation of existing buildings. Assessing the carbon footprint during the life cycle of buildings using LCA methodology should take precedence over the energy efficiency.

The higher energy consumption of masonry residential buildings is not only due to the inadequate thermal insulation properties of the masonry external walls, but also to the often inadequate properties of casement and dual-timber single-glazed windows. Improving the thermal insulation properties of windows, using new infill structures, and the effective refurbishment and repair of existing openings can make a significant contribution to improving the energy performance of these buildings (Figure 9). However, when addressing this issue, it is necessary to consider not only the properties of the windows themselves, but also the related structures [25–27]. The technical parameters of window infills that need to be evaluated mainly include the heat transfer coefficient, airtightness, air permeability and watertightness.

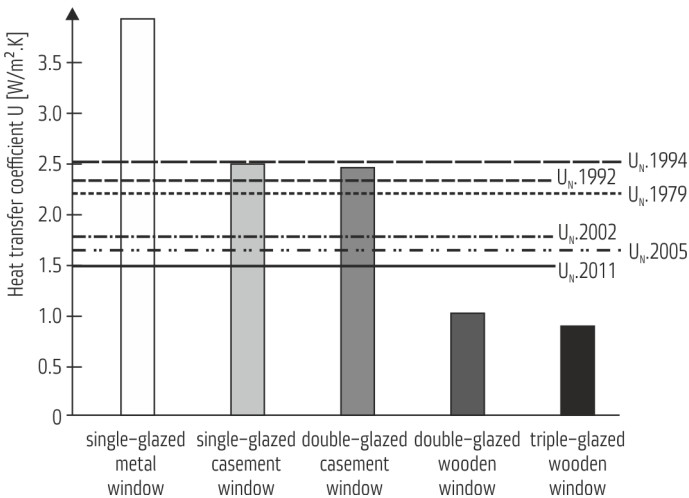

**Figure 9.** Comparison of heat transfer coefficient for different types of windows and glazing. (Analysis by the authors).

Achieving optimum thermal properties, the comfort and healthiness of the indoor environment, the prevention of mould formation and the reduction in energy consumption of masonry residential buildings requires a number of complex and systemic measures. External insulation of envelope masonry walls with a contact insulation system cannot be applied to masonry residential buildings with the decorative design of a segmented street façade with a number of cornices, balconies and bay windows. Contact insulation systems on the outside can only be applied in cases that allow the original simpler profiles to be preserved, especially for courtyard facades.

A number of problems that have not yet been sufficiently addressed accompany the insulation of the articulated façade of the street envelope walls from the inside. The severity of these problems increases with the increasing humidity of the insulated interior. For internal wall insulation, thermal insulations with high diffusion resistance such as foam glass and extruded polystyrene are preferred. The combination of moderate internal insulation with external thermal insulation plaster will reduce the amount of water vapour condensing in the structure in winter to a certain extent. However, the rugged and intricately profiled street façade of the historicizing architecture of these buildings does not allow the application of an insulating plaster. A vapour barrier on the inside of the structure is usually proposed as a solution. Despite several measures, the vapour barrier cannot be considered perfectly tight; the risk of various leaks is always high even with very careful implementation. In the analysis of the proposed structure, the diffusion resistance factor of the vapour barrier must always be reduced to at least one-tenth of the value stated by the

manufacturer. Due to the many uncertainties in the implementation of the vapour barrier, thermal insulation materials with high diffusion resistance are preferred for wall insulation. One of the most common alternative methods of internal insulation is the use of so-called capillary active materials, which allow the transport of moisture in the liquid phase. In contrast to the traditional solution, in which the vapour barrier restricts the diffusion of water vapour into the structure, with capillary active materials, the condensation of water vapour is accepted as an inevitable process—the condensate formed is dispersed in the capillary active insulation and evaporates into the interior when the boundary conditions change. A serious problem in these insulation cases is the slight increase in the mass moisture content of the insulated masonry (various international studies report an average increase of 0.5 to 1.5 percentage points). This increase from the pre-insulation condition is usually higher the greater the thickness of insulation applied and the thinner the insulated wall [22]. Finally, the internal insulation also reduces the internal space of the building.

Structures that can also adversely affect the energy balance of the building and whose additional insulation is necessary from the point of view of energy savings are ceiling structures above the unheated basement space and ceiling structures under the pitched roof structure, or ceiling structures under the unheated attic.

By insulating the ceiling structures under the attic and above the unheated basement, the temperature in the attic or basement will drop in winter. The reduction in air temperatures can then cause condensation on the cold roof trusses or basement walls. From this point of view, it is necessary to assess the possible consequences of these measures in relation to the risk of condensation on the structures where the temperature has fallen as a result of the insulation. Restrictions on the original ventilation of the attic may pose a moisture risk to the timber roof structure.

Compared to envelope structures, the benefit gained by insulating the ceiling structure under a pitched roof structure is not as significant, as the temperature gradient of that structure is significantly lower. The losses of a brick apartment building in its original state through the envelope (without windows) account for 40–65% of the building losses; losses through the ceiling under the unheated attic 14–20%; losses through the floor above the unheated basement about 10–15%; and losses through the original windows about 5–15%.

When modifying wooden ceilings for the purpose of thermal insulation or the improvement of acoustic properties, it is necessary to avoid layers with high diffusion resistance (PVC, rubber, cardboard, foil, etc.), which could change the existing hygrothermal regime of the ceiling.

Particular attention should be paid to the placement of the ceiling joists in the pockets in the perimeter (outside) walls. Preventing beam ceiling header degradation requires ensuring the necessary warm air flow on the inner surface of the perimeter walls, limiting damp operations (ensuring effective ventilation) and, finally, ensuring the permeability of the beam ceiling structure.

Thermal comfort and energy losses are also influenced by the thermal properties of internal load—bearing and dividing walls, and partitions. According to CSN 73 0540-2: 2011, the heat transfer coefficient of these structures must be lower than 0.60 W/(m$^2$.K) or 0.75 W/(m$^2$.K), depending on whether they separate the heated space from the tempered space (e.g., from an internal corridor in an apartment building) or from a completely unheated space (e.g., from a garage or basement). Existing internal wall structures do not have the necessary thermal insulation properties.

A number of research institutes, especially in Central and Northern Europe, are dealing with the restoration, modernisation and insulation of historic buildings. Theoretical and experimental research works carried out until now have not yet provided a reliable and unambiguous answer to the questions related to the insulation of envelope structures from the interior side. Partial results and conclusions are mainly linked to specific (local) conditions and materials and cannot be directly and reliably applied to the insulation of the articulated and profiled facades of historic buildings.

The extent and importance of improving the performance and functional properties of masonry envelope walls and associated structures requires the design of a reliable and durable solution supported by both theoretical and experimental research.

The literature mostly deals with only partial, specific problems. Quite extensive research is being conducted in Denmark, Slovenia, the Baltic countries, and outside Europe in the USA, Canada and China. The research mainly focuses on testing different types of thermal insulation (porous concrete, calcium silicate boards and capillary active thermal insulation system based on rigid PUR foam panels) bonded in different thicknesses to the inside of brick walls [28], as well as different thicknesses and extents of insulation (e.g., parapet masonry only) [29,30]. Furthermore, the risk of changing temperature and humidity conditions inside the original wall (increase in mass moisture due to condensation) [31–34], the associated negative effect of freezing cycles [35] and the risk of mould growth [36–38] have been investigated. The possible reduction in the influence of these risks by using capillary active materials in combination with the installation of heating cables is suggested in [39].

The analysis of different structural solutions (internal insulation with and without vapour barrier, with and without hydrophobicity, etc.) highlights problematic locations in the area of the installation of wooden ceiling structures (increased moisture impact, reduced temperatures in the installation area, etc.) [38,40–42].

Furthermore, a part of the research is focused on the modelling and numerical simulations of the thermal and moisture behaviour of masonry structures while using different types of insulation [43,44].

In the case of changes and structural modifications of completed buildings, according to CSN 73 0540-2: 2011 (Article 5.2.11), it is possible to exceed the required values of the heat transfer coefficient if technical or legislative obstacles demonstrably prevent compliance with the standard's requirement. However, in such a case, at least the best technically available solution must be used so that defects and malfunctions in the use of the building are demonstrably prevented.

### 2.3. Hygrothermal Analysis of Selected Structures and Details

In historic brick buildings, wooden (timber) beam (ceiling) structures were used to a large extent for the roofing of interior spaces until the mid-19th century. The most common type of the vertical load-bearing structure was a masonry wall 450 mm thick, made from burnt bricks and cement-lime mortar. Where the timber beams are placed on the external load-bearing wall (in the so-called pockets), the external wall is weakened by 200 to 250 mm, and thus, the thermal resistance is significantly reduced (Figure 10a). A similar situation occurs in the case of ceramic ceilings (Figure 10b). Therefore, a two-dimensional steady state hygrothermal analysis of this critical detail in the original state (without any thermal insulation) was performed for the case of wooded as well as ceramic ceiling. To evaluate possible solutions, an analysis of the structure with different positions of thermal insulation (external and internal with or without vapour barrier) was also conducted (see Figure 10).

As previously mentioned, in the case of residential buildings built at the turn of the 19th and 20th centuries in the Central European region, it is often not possible to apply external thermal insulation due to the decorative design of street façades. Hence, the internal thermal insulation is often the only viable solution. However, when applying internal insulation, thermal bridges are created around the wall contacts with the surrounding structures, especially with partitions, walls, ceilings and roofs (Figure 11). In buildings with timber beam ceiling structures, internal insulation increases the risk of condensation around the beam head (see Figure 10). A two-dimensional steady state hygrothermal analysis of a selected section of the historic structure containing an external wall and inner partition contact and external wall corner was performed to compare the impact of thermal insulation placement (external insulation composed of expanded polystyrene vs. internal insulation composed of capillary active material—polyurethane foam).

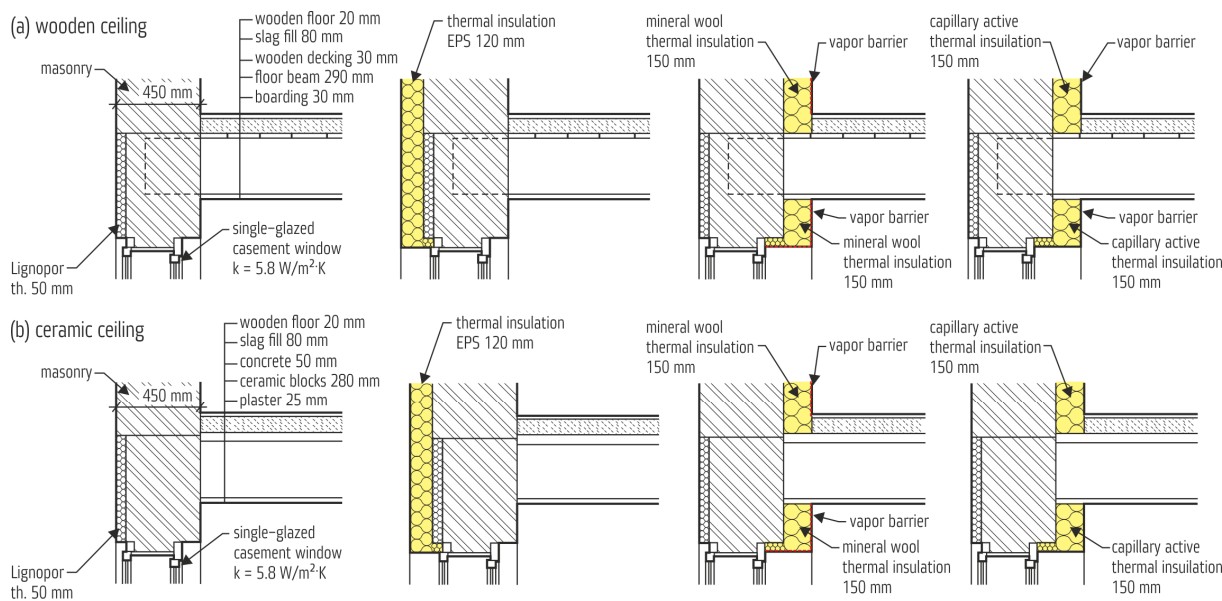

**Figure 10.** Variants of models for hygrothermal analysis: (**a**) external wall with window lintel and wooden ceiling structure—without insulation, with external insulation, with internal insulation and vapour barrier and with capillary active internal insulation; (**b**) external wall with window lintel and ceramic ceiling structure—without insulation, with external insulation, with internal insulation and vapour barrier and with capillary active internal insulation. (Analysis by the authors).

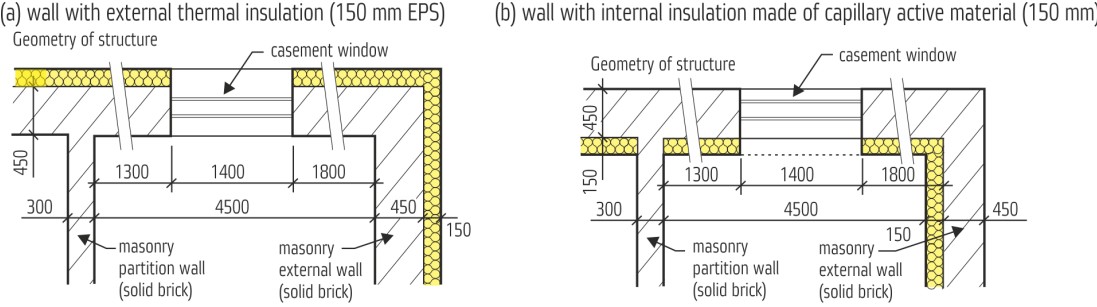

**Figure 11.** Variants of models for hygrothermal analysis: (**a**) external wall with external insulation; (**b**) external wall with capillary active internal insulation. (Analysis by the authors).

Finally, the subject of dual windows with single glazing in historic masonry structures was investigated. The improvement of their thermal insulation properties is often limited by the number of boundary conditions. Double widows in historic structures can be renovated by replacing both sashes with new double- or triple-glazed windows, or (in the case of historically valuable buildings) by replacing only the outer sashes with new ones with high-quality double or triple glazing. Double glazing (or triple glazing) with a low heat transfer coefficient fitted in the outside position will ensure sufficiently high internal surface temperatures both on the sashes of the dual window and on the casing or plastered part of the lining between them. Internal surface temperatures on the surrounding wall may also increase significantly. This minimizes the risk of condensation on the inner surface of all components of the dual window. On the other hand, placing the double (or triple) glazing in the inner sashes of the dual window can lead to a drop in air temperature in the cavity between the sashes in winter and can cause condensation and icing of the condensate on the inner surface of the outer single-glazed sash. Surface condensation can also be a threat to the casing if not effectively insulated. Therefore, a comparison of these possible solutions was carried out in the form of a two-dimensional steady state hygrothermal analysis (Figure 12).

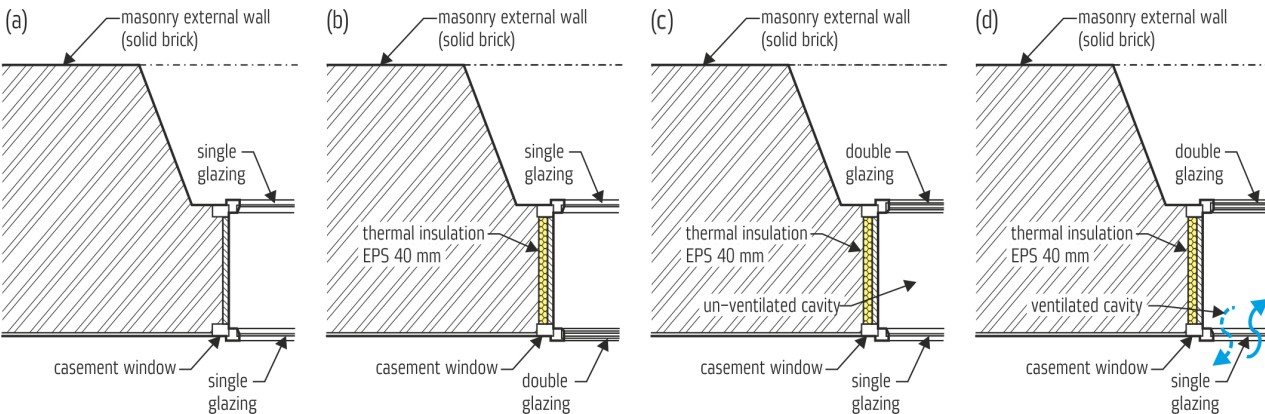

**Figure 12.** Variants of models for hygrothermal analysis: (**a**) original dual wooden window, plastered lining without thermal insulation; (**b**) modified dual window with double glazing in the outer sash, insulated lining with plaster; (**c**) modified dual window with double glazing in the inner sash and with unventilated cavity; (**d**) modified dual window with double glazing in the inner sash and with ventilated cavity.

Numerical Two-Dimensional Steady State Hygrothermal Analysis

In all the two-dimensional steady state hygrothermal analyses (two-dimensional heat conduction) performed, standard material properties (particularly coefficient of thermal conductivity) were used—masonry $\lambda = 0.8$ W.m$^{-1}$.K$^{-1}$; wood $\lambda = 0.18$ W.m$^{-1}$.K$^{-1}$; slag fill $\lambda = 0.95$ W.m$^{-1}$.K$^{-1}$; lime plaster $\lambda = 0.88$ W.m$^{-1}$.K$^{-1}$; expanded polystyrene (EPS) $\lambda = 0.039$ W.m$^{-1}$.K$^{-1}$; mineral wool $\lambda = 0.041$ W.m$^{-1}$.K$^{-1}$; polyurethane foam $\lambda = 0.031$ W.m$^{-1}$.K$^{-1}$; Lignopor $\lambda = 0.046$ W.m$^{-1}$.K$^{-1}$; glass $\lambda = 0.78$ W.m$^{-1}$.K$^{-1}$; closed air gap $\lambda = 0.067$ W.m$^{-1}$.K$^{-1}$; wooden window frame $\lambda = 0.13$ W.m$^{-1}$.K$^{-1}$.

Standard boundary conditions pursuant to the CSN 73 0540-2: 2011 were considered (indoor air temperature 22 °C, indoor relative humidity 50%, outdoor air temperature −13 °C, outdoor relative humidity 84%).

The software used to perform the two-dimensional steady state hygrothermal analysis was Area 2017 by Svoboda software. This software calculates temperature and pressure fields solving standard partial differential equation [45]:

$$\frac{\partial}{\partial x}\left[\lambda(x,y)\left[\frac{\partial\Theta(x,y)}{\partial x}\right]\right] + \frac{\partial}{\partial y}\left[\lambda(x,y)\left[\frac{\partial\Theta(x,y)}{\partial y}\right]\right] = 0 \qquad (1)$$

with the boundary condition:

$$-\lambda(x,y)\left[\frac{\partial\Theta(x,y)}{\partial n}\right] = h\left(\Theta - \overline{\Theta}\right) \qquad (2)$$

where (x,y) are the coordinates of the point in the plane [m]; $\lambda$ is the thermal conductivity coefficient [W/(m.K)]; $\theta$ is the temperature at the point [°C]; h is the heat transfer coefficient [W/(m$^2$K)]; $\theta$ is the ambient temperature [°C]; $\partial x$, $\partial y$ are the derivatives with respect to x and y; and $\partial n$ is the derivative with respect to the normal.

Equation (1) is solved on a simply continuous region $\Omega$ with boundary $\Gamma$, on which the boundary condition (2) must be satisfied. The boundary $\Omega$ is rectangular or generally curvilinear. Furthermore, it is assumed for the calculation that the region $\Omega$ can be divided into a finite number of regions in which the function $\lambda(x,y)$ is constant. Moreover, the functions h(x, y) and $\theta(x,y)$ are considered constant over parts of the boundary $\Gamma$. Equation (1) is modified by the Galerkin method and using Green's theorem to the form:

$$K \cdot r = q \qquad (3)$$

where K is the body conductivity matrix defined as

$$K = \int_{\Omega} \left[ \lambda \frac{\partial N}{\partial x} \cdot \frac{\partial N^T}{\partial x} + \lambda \frac{\partial N}{\partial y} \cdot \frac{\partial N^T}{\partial y} \right] d\Omega \tag{4}$$

and r is the column matrix of nodal temperature values (unknowns) and q is the right-hand side vector (source vector) described by

$$q = \int_{\Gamma} N \cdot h \cdot \left( \overline{\Theta} - N^T \cdot r \right) d\Gamma \tag{5}$$

and N is a row matrix of basic functions.

Discretization of the problem (generation of finite elements into which the region G is divided) is performed using a planar triangular element. The final solution of the system of linear equations (calculation of the temperature and pressure values in the nodes) is performed using Gaussian elimination.

For the calculation of the two-dimensional stationary field of water vapor pressures, a similar equation and boundary condition is used—only instead of the thermal conductivity coefficient, it is necessary to consider the diffusion coefficient, and instead of the heat transfer coefficient, the water vapor transfer coefficient.

## 3. Discussion of Results

The results of the hygrothermal analysis performed on the non-insulated section of a historic masonry structure with a wooden ceiling showed a significant decrease in the temperature in the place of the wooden beam placement, compared to the normal part of the external wall. The temperature around the beam head varied between 0 °C and +8 °C, which is below the dew point temperature (around +10 °C for the considered boundary conditions). This means a high risk of water condensation both in the masonry structure and on the contact surface between the masonry and the wooden beam. In the case of the ceiling with ceramic liners (first half of the 20th century), water vapour condensation occurred on the inside surface of both the wall and the bond beam at an outside design temperature of −13 °C. In both cases, water condensation also occurred on the outer sash of the wooden dual window (Figure 13).

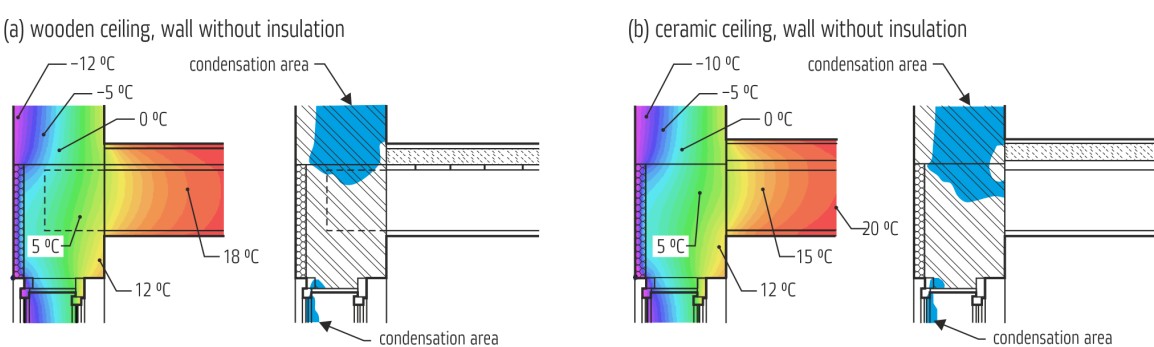

**Figure 13.** Results of hygrothermal analysis: (**a**) external wall with window lintel and wooden ceiling structure for the variant without insulation; (**b**) external wall with window lintel and ceramic ceiling structure for the variant without insulation. (Analysis by the authors.) By insulating the structure with an external insulation system, the thermal and humidity regime of the external masonry wall is significantly improved, the overall thermal comfort of the internal environment is improved, and the energy consumption is reduced (Figure 14). Water vapour condensation is limited to the usual point at the interface between the thermal insulation and the thin plaster; no surface condensation occurs. Again, a small amount of water condensation occurs on the outer sash of the wooden dual window. (Analysis by the authors).

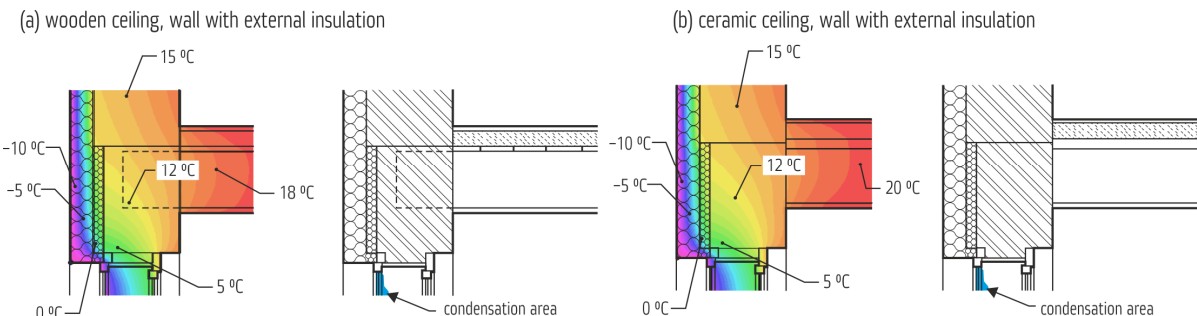

**Figure 14.** Results of hygrothermal analysis: (**a**) external walls with window lintel and wooden ceiling structure with external insulation; (**b**) external walls with window lintel and ceramic ceiling structure for the variant with external insulation. (Analysis by the authors).

In the case of internal thermal insulation, two different models were studied. The first one utilizes thermal insulation composed of mineral wool together with vapour barrier on the inside of the structure. In this case, for both variants of ceiling structure (wooden and ceramic), the temperature drop was even more significant than for the non-insulated structure (the external wall structure is effectively insulated from the indoor source of heat), so the appropriate conditions for water condensation appeared in a wider area of the historic structure. However, due to the vapour barrier, the humidity donation from the interior to the structure was limited. In both cases, there was a problematic spot in the place of the external wall and ceiling structure connection, which presented a discontinuity in the vapour barrier, allowing some moisture access into the structure. The water condensation area was bigger in the case of the ceramic ceiling due to the material properties of the ceiling structure (Figure 15).

The second variant of internal thermal insulation utilizes capillary active material (polyurethane foam) with no vapour barrier. Again, for both variants of the ceiling structure, the temperature drop was more significant than for the non-insulated structure. Moreover, due to the lack of vapour barrier, water condensation occurred in wider areas of the structure. More significant condensation occurred in the case of the ceramic ceiling structure. However, due to the active porous system of the thermal insulation used, the condensed water could again evaporate into the indoor environment, once the conditions were favourable (reversed heat flow when the outside temperature exceeded the indoor temperature). Nevertheless, the period of time that the condensed water is in the structure can be quite significant and can cause deterioration of the structure, impair its properties and worsen if the indoor air quality decreases due to the mould growth inside the thermal insulation and on its surface.

The results of the hygrothermal analysis performed on the section of historic structure in the places of external and internal wall connection and in the place of the external wall corner showed that thermal bridges in internal insulation, compared to the uninterrupted thermal insulation envelope in the case of external insulation, cause higher heat losses through penetration and the overall degradation of insulation performance. Moreover, there is a possible water condensation in this case, which is not present in the case of external insulation (Figure 16).

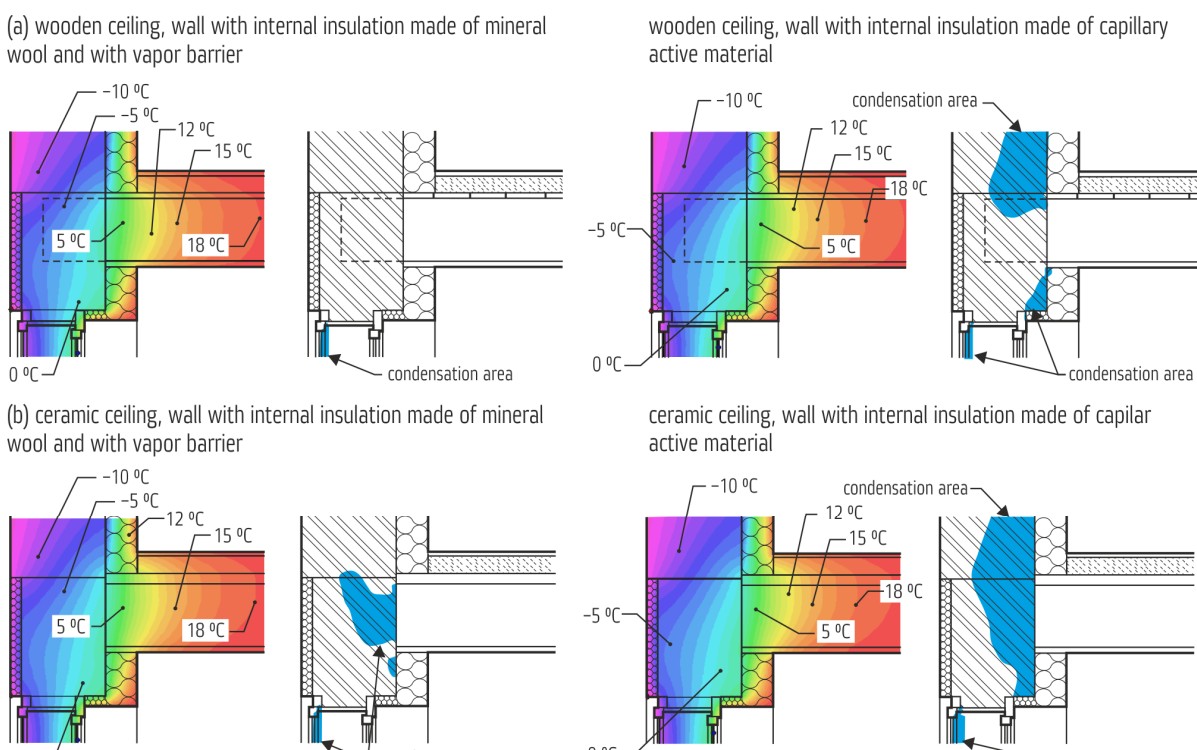

**Figure 15.** Results of hygrothermal analysis: (**a**) external walls with window lintel and wooden ceiling structure for the variant with internal insulation composed of mineral fibre and vapour barrier, and internal insulation with capillary active material; (**b**) exterior walls with window lintel and ceramic ceiling structure for the variant with internal mineral fibre insulation and vapour barrier, and internal insulation with capillary active material. (Analysis by the authors).

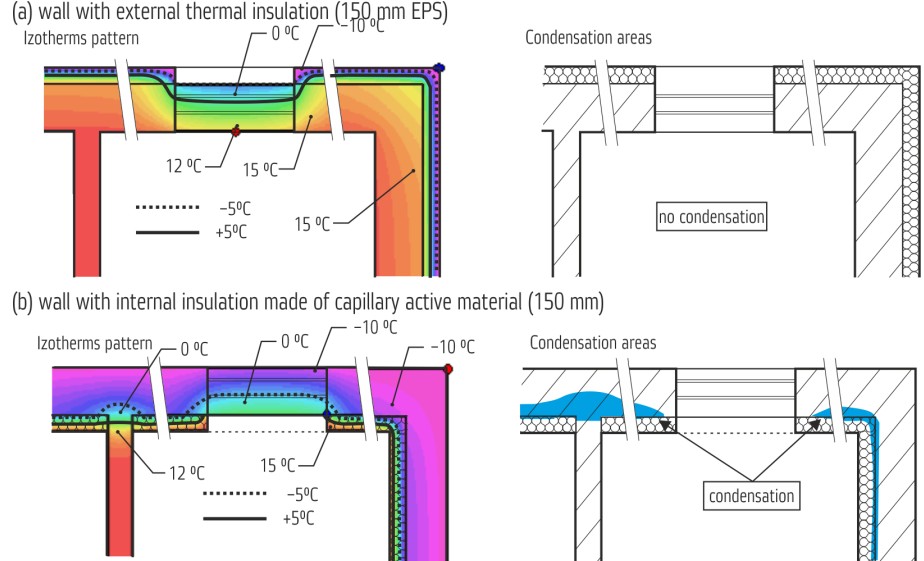

**Figure 16.** Results of hygrothermal analysis: (**a**) external walls with thermal insulation composed of EPS 150 mm thick on the external side; (**b**) external walls with thermal insulation composed of capillary active material 150 mm thick on the internal side. (Analysis by the authors).

A comparison of the isotherms' pattern and the size of the condensation zones for the different dual window solutions is shown in Figure 17. It is clear from the graphical outputs that water vapour condensation occurs on the inner surface of the outer single

glazing at very low outside temperatures, even in the case of a traditional dual window solution (Figure 17a). The dual window solution with double glazing in the inner frame results to be very similar—if very tight new frames were used for the inner glazing, the rate of water vapour condensation could even be slightly lower (Figure 17c). Clearly, the most favourable results can be obtained for a solution with double glazing in the outer sash, which should be designed and implemented as standard (Figure 17b).

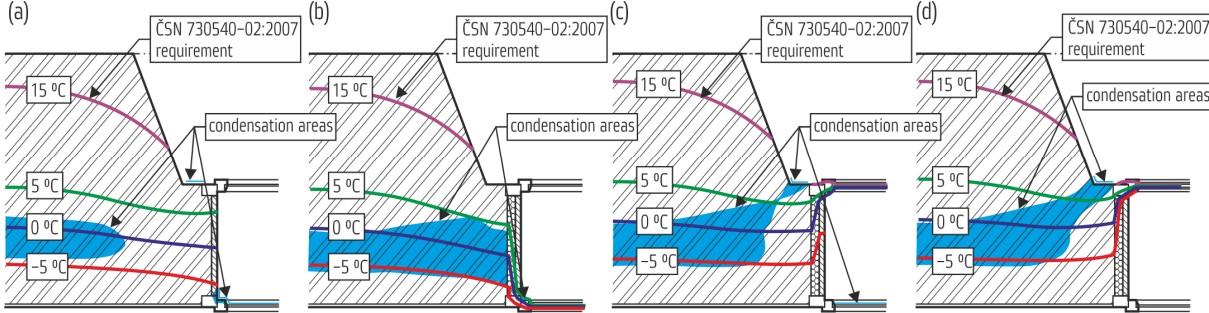

**Figure 17.** Results of hygrothermal analysis: (**a**) original dual wooden window, plastered lining without thermal insulation; (**b**) modified dual window with double glazing in the outer sash, insulated lining with plaster; (**c**) modified dual window with double glazing in the inner sash, with unventilated cavity; (**d**) modified dual window with double glazing in the inner sash, with ventilated cavity. (Analysis by the authors).

However, when implementing the otherwise inappropriate double (or triple) glazing in internal sashes, a number of additional and not always simple measures are required. Such a window structure should be designed as a double (double-skin) façade. This means that in such a window, only the inner part with double (or triple) glazing will provide thermal protection, while the outer sashes with single glazing will mainly have an acoustic and protective function (against climatic influences). The air cavity between the inner and outer sashes must be sufficiently ventilated with outside air, so that water vapour can be removed before it condenses or freezes on the cold outer glazing. Ensuring ventilation requires making openings for air inlet and outlet; it is insufficient to simply omit the seals in the outer sashes, for example. In more complex situations, it is necessary to perform a detailed computational verification of the design of ventilation openings using an CFD analysis. It is also necessary to design the lining insulation and to verify it computationally. It is not always possible to place thermal insulation of sufficient thickness between the two sub-windows (sashes) to completely eliminate surface condensation (Figure 17d). With careful design, such a modified dual window may perform without major problems in terms of humidity, but it will always be inferior to a dual window with double glazing in the outer sashes in terms of overall thermal insulation performance.

## 4. Conclusions

More than 70% of the building stock needed for the development of society in the 21st century has already been built. This large stock needs to be maintained and adapted to new conditions and requirements. The renovation and maintenance of the existing building stock should be the subject of increased research interest worldwide. Research should primarily focus on the objective assessment of the structural and technical condition and residual life of buildings, addressing the increasing requirements for the reliability, functionality, energy efficiency and healthiness of buildings; environmental impact; recycling of building materials, etc. The issues of defects and failures, degradation processes and the design of building reconstruction cover a wide range of issues of an interdisciplinary nature, from the natural sciences, materials engineering, mechanics, flexibility and statics of building structures, to knowledge of historical materials, structures and technologies.

Knowledge of historic structures, materials and construction methods used can prevent errors in the restoration of historically significant, especially listed, buildings.

The renovation and modernisation of these brick tenement buildings is a means of preserving them and making them fully functional in the future. It is a prerequisite that this restoration and modernisation fully preserves their architectural and historic value and character; achieves acceptable compliance with the requirements of the provisions of the regulations and standards currently in force; and demonstrates cost-effectiveness. It can be assumed that in the near future, these parts of historic towns, documenting their often "turbulent" development in the late 19th and early 20th centuries that began with the Industrial Revolution, will become part of heritage-protected urban development. Neglecting the restoration, reconstruction and modernisation of these residential buildings in order to achieve adequate housing quality and sustainability would be the beginning of the "dilapidation" of these buildings and their gradual abandonment by the inhabitants, and thus, the creation of socially problematic urban districts.

This paper analyses the issue of the structural and physical properties of vertical load-bearing structures and their infills, which must be addressed in the context of urban tenement houses from the second half of the 19th and early 20th century. As a rule, these buildings do not achieve the quality of the internal environment according to today's standards and are energy-inefficient. At the same time, the greatest potential for changing energy consumption lies in these already built buildings, as they consume up to 40% of all energy produced and produce up to 36% of emissions. Therefore, renovating existing buildings to an energy-efficient standard can reduce the energy demand of buildings by up to 70%. The functional revitalisation of these buildings, taking into account their energy performance, should be a part of sustainable development for these reasons and could also make a significant contribution to meeting the Kyoto targets in line with the EU Energy RoadMap 2050.

**Author Contributions:** Conceptualization, methodology, formal analysis, investigation, resources; writing—original draft preparation, writing—review and editing, supervision, project administration, funding acquisition K.K. Methodology, software, validation, writing—review and editing, visualization R.Z. All authors have read and agreed to the published version of the manuscript.

**Funding:** This research was funded by Ministry of Culture of Czech Republic, grant number NAKI DG18P02OVV038.

**Data Availability Statement:** Some or all data, models, or code generated or used during the study are available from the corresponding author by request (FEM numerical models and results of numerical analyses).

**Acknowledgments:** This article was written as part of the NAKI DG18P02OVV038 research project "Traditional City Building Engineering and Crafts at the Turn of 19th and 20th Centuries (2018–2022, MK0/DG)".

**Conflicts of Interest:** The authors declare no conflict of interest.

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
