# Peer review of "Building and Architectural Details of Tenement Houses Built at the Turn of the 19th and 20th Centuries in Central European Region—Hygrothermal Analysis"

_buildings, doi:10.3390/buildings13020451_

Round 1

Reviewer 1 Report

The topic of the study is interesting and important; it provides a good historical background to reason the main current problems in building envelopes from the mid-19-20th cent. in the Czech lands to provide recommendations to overcome them. However, some revisions must be undertaken prior to publishing.

The work is focused on buildings in the Czech lands; therefore, I suggest modifying the title to reflect the focus of the study.

The main issue I found in the paper regards the lack of discussion on the cultural heritage status of an important part of the building stock of this period and how the new regulations and recommendations for energy efficiency can be adapted on these; the word “Heritage” is only mentioned in the concluding section of the paper.

I suggest including the following reference for the discussion:

Troi A. Recommendations for Local Governments: Integrating energy efficient retrofit of historic buildings into urban sustainability 2010:1–52.

I also miss some commentary on the implementation of the new EU directives: The energy performance and upgrading of buildings was introduced in the Czech legislation in the first decade of 2000. Since 2009, it is required for certain types of buildings to reach a specific levels of energy efficiency. These requirements are getting stricter, following the implementation of new EU Directives. I think that the paper would benefit from including a discussion on this.

Figures 3 and 4 - I suggest providing larger images.

Author Response

Dear Reviewer,

here´re the answers and comments to your review:

  • The title of the article was changed to "Building and architectural details of tenement houses built at the turn of the 19th and 20th centuries in Central European region – Hygrothermal analysis" so it reflects the focus of the study.
  • We have expanded the section in the article focusing on recommendations and options related to increasing the energy efficiency of apartment buildings, including an expansion of the literature used.
  • A short information about the EU directives was added, however, deeper discussion is out of the scope of presented paper.
  • The size of the images is at the publisher's discretion, their resolution is sufficient for possible enlargement.

With best regards,

Klara Kroftová

Reviewer 2 Report

The topic of the article is absolutely in line with all global strategies in this area. In Europe, for example, it is the New European Bauhaus initiative. In connection with the city, we can talk about urban recycling. When its subject is not the recycling of building materials, but the recycling of the urban structure itself. In the case of energy efficiency, it is not primarily an individual object, but an entire urban fragment. Preserving the cultural integrity of entire parts of the city and the city as a whole is crucial. Assessing the carbon footprint during the life cycle of buildings should take precedence over primary energy efficiency. The partial researches presented in the article are indirectly aimed at this goal. A clearer structuring of the article and a more comprehensive view are missing.

Line 20, why carbon footprint is not taken into account? Line 31, please look for a synonym for "construction".

Figure 1., original picture by the authors, or by permission by...?

Figure 2., is a reference for using a picture enough?, or a permission of the copyright is needed?, the editor has to decide. The same goes for several of the following images. It should be stated..., by the authors, or another source in a qualified manner.

Line 130, architectural expression?

Line 135, in my opinion, the cultural value in the exposed positions takes precedence over the thermal quality of the building.

Line 178, it is necessary to explain the abbreviation CSN (!)

Figures 6., 7., 8., as mentioned before, by the authors, or source Line 215, why EPBD is not directly mentioned if we are talking about the current state.

Line 218, the statement is self-explanatory.

Figure 10. and the following, the simulation method has been known for years, the use is OK, only again..., analysis done by the authors.

Line 320. Carbon footprint calculations using the LCA methodology are known for sensibly refurbished historic buildings. The operational energy requirement is higher, but the carbon footprint is reduced by the fact that the bearing structure already exists. I refer e.g. for the project Monumentum ad usum DUK Krems.

Line 342, a certain disadvantage of using internal thermal insulations is also the reduction of internal space.

The article is essentially a summary of existing knowledge. After partial analyses, a case study at the level of the reconstruction project of one rental house with overall calculations, or an analytical assessment of one completed reconstruction, is missing. We can find enough examples in Prague. A high-quality case study could be a decisive contribution of the submitted article.

Author Response

Dear Reviewer,

here´re the answers and comments to your review:

The topic of the article is absolutely in line with all global strategies in this area. In Europe, for example, it is the New European Bauhaus initiative. In connection with the city, we can talk about urban recycling. When its subject is not the recycling of building materials, but the recycling of the urban structure itself. In the case of energy efficiency, it is not primarily an individual object, but an entire urban fragment. Preserving the cultural integrity of entire parts of the city and the city as a whole is crucial. Assessing the carbon footprint during the life cycle of buildings should take precedence over primary energy efficiency. The partial researches presented in the article are indirectly aimed at this goal. A clearer structuring of the article and a more comprehensive view are missing.

- The authors considered the reviewers valuable comments and made extensive modifications to the article, namely the structure of the article was completely changed

Line 20, why carbon footprint is not taken into account? Line 31, please look for a synonym for "construction".

- Thank you for the suggestion, added to text. However, the carbon footprint analysis was out of the scope of this article

Figure 1., Figure 2.

- All the pictures are drawn/taken by authors, or the permission is not needed - they´re public domain, the authors are more then 70 years dead.

Line 130, architectural expression?

- Thank you for the suggestion, text changed.

Line 135, in my opinion, the cultural value in the exposed positions takes precedence over the thermal quality of the building.

- Thank you for the suggestion, text changed.

Line 178, it is necessary to explain the abbreviation CSN (!)

- abbreviation explained and CSN listed in references.

Figures 6., 7., 8., as mentioned before, by the authors, or source Line 215, why EPBD is not directly mentioned if we are talking about the current state.

- Figures description amended, EPBD mentioned

Line 218, the statement is self-explanatory.

- OK

Figure 10. and the following, the simulation method has been known for years, the use is OK, only again..., analysis done by the authors.

- figures description amended

Line 320. Carbon footprint calculations using the LCA methodology are known for sensibly refurbished historic buildings. The operational energy requirement is higher, but the carbon footprint is reduced by the fact that the bearing structure already exists. I refer e.g. for the project Monumentum ad usum DUK Krems.

- very interesting suggestion, However, the LCA is out of the scope of current article

Line 342, a certain disadvantage of using internal thermal insulations is also the reduction of internal space.

- thank you for the suggestion, added to text

The article is essentially a summary of existing knowledge. After partial analyses, a case study at the level of the reconstruction project of one rental house with overall calculations, or an analytical assessment of one completed reconstruction, is missing. We can find enough examples in Prague. A high-quality case study could be a decisive contribution of the submitted article.

- very interesting suggestion, However, a case study is out of the scope of current article.

Best regards,

Klara Kroftova

Reviewer 3 Report

The manuscript addresses a very compelling and topical issue: building physics issues of tenement houses built at the turn of the 19th and 20th centuries. Unfortunately, the manuscript is not written in an academic manner, lacking the basic elements of a scientific text: an adequate introduction with a clearly stated research goal and a review of existing research on the same topic. The text under the Materials and Methods chapter does not meet the specified topic of the chapter. The research methodology is not presented, it is not known how the research was conducted. I assume that the chapter 3. Building Physics should be the chapter where you present the results of your research. From the text in the mentioned chapter, it is not entirely clear whether you have obtained the results based on the researched literature or based on your own research. In the first case, the facts are not sufficiently supported by literature sources, and in the second case, it is not clear how you reached certain conclusions. Figures 1 (line 50), 6 (line 192), 7 (line 195), 8 (line 220), 9 (line 230), 10 (line 242), 11 (line 311), 12 (line 378), 13 (line 385) and 14 (line 411) do not list the sources on which specific diagrams or drawings were based. Not all references are listed under bibliography, for example: CSN 73 0540. The manuscript lacks a discussion chapter in which you discuss the obtained results and interpret them in relation to previous research. The final chapter (4. Conclusions) contains a well addressed and elaborated research problem, but none of the solutions were offered. Before uploading the manuscript, the Buildings template should have been studied, and the research results should have been written according to the instructions for the authors.

Author Response

Dear Reviewer,

please find bellow the reactions and comments to your review:

The manuscript addresses a very compelling and topical issue: building physics issues of tenement houses built at the turn of the 19th and 20th centuries. Unfortunately, the manuscript is not written in an academic manner, lacking the basic elements of a scientific text: an adequate introduction with a clearly stated research goal and a review of existing research on the same topic. The text under the Materials and Methods chapter does not meet the specified topic of the chapter. The research methodology is not presented, it is not known how the research was conducted. I assume that the chapter 3. Building Physics should be the chapter where you present the results of your research. From the text in the mentioned chapter, it is not entirely clear whether you have obtained the results based on the researched literature or based on your own research. In the first case, the facts are not sufficiently supported by literature sources, and in the second case, it is not clear how you reached certain conclusions. Figures 1 (line 50), 6 (line 192), 7 (line 195), 8 (line 220), 9 (line 230), 10 (line 242), 11 (line 311), 12 (line 378), 13 (line 385) and 14 (line 411) do not list the sources on which specific diagrams or drawings were based. Not all references are listed under bibliography, for example: CSN 73 0540. The manuscript lacks a discussion chapter in which you discuss the obtained results and interpret them in relation to previous research. The final chapter (4. Conclusions) contains a well addressed and elaborated research problem, but none of the solutions were offered. Before uploading the manuscript, the Buildings template should have been studied, and the research results should have been written according to the instructions for the authors.

  • The authors considered the reviewers valuable comments and made extensive modifications to the article, namely the structure of the article was completely changed.
  • The sources of figures sources have been modified and clarified. Also the list of references has been expanded and missing sources have been added.

With best regards,

Klara Kroftova

Reviewer 4 Report

The paper presents the hydrothermal analysis of solutions used in tenement houses from the 19th and 20th centuries. The study is interesting and brings some contributions. Below are some comments that may improve the presentation of the article.

- The title says "Building physics issues," but I think "physics" is an extensive term, including several areas. I suggest making it more specific with the analysis that was carried out. Something like "Hygrothermal analysis". The usage could also be reviewed throughout the text.

- Section 2: Material and Methods. I don't think this section describes the materials and methods, but it does describe the structure types. As a suggestion, it could be something like "Building Structure".

- A Materials and Methods section should present the tools used, with information so the work can be reproduced. I thought it was missing information about how the hydrothermal analysis was done.

- I suggest including the cited norms in the references so that it is possible to find them.

- The analyzes were carried out in a permanent regime. Would it be important to consider the dynamics of the structure? Its behavior over time.

- I suggest discussing the results obtained against other similar scientific works, such as those presented throughout the text.

Author Response

Dear Reviewer,

please find bellow the reactions and comments to your review:

- The title says "Building physics issues," but I think "physics" is an extensive term, including several areas. I suggest making it more specific with the analysis that was carried out. Something like "Hygrothermal analysis". The usage could also be reviewed throughout the text.

- Section 2: Material and Methods. I don't think this section describes the materials and methods, but it does describe the structure types. As a suggestion, it could be something like "Building Structure".

- The title of the article and sections have been changed.

- A Materials and Methods section should present the tools used, with information so the work can be reproduced. I thought it was missing information about how the hydrothermal analysis was done.

- Thank you for the suggestion, the text has been modified accordingly.

- I suggest including the cited norms in the references so that it is possible to find them.

- The list of literature has been expanded and missing sources have been added.

- The analyzes were carried out in a permanent regime. Would it be important to consider the dynamics of the structure? Its behavior over time.

- Thank you for the suggestion. Unfortunately, the software available to the authors allows only stable hygrothermal analysis.

- I suggest discussing the results obtained against other similar scientific works, such as those presented throughout the text.

- Thank you for the suggestion, deeper discussion of the results (current and other in preparations) against other works is planned in a follow-up article.

With best regards,

Klara Kroftova

Round 2

Reviewer 1 Report

The authors have extensively revised the manuscript, and its quality has improved. There are just a few points still require revision:

- Line 68: I suggest substituting “not cultural monuments” with “not classified as cultural monuments” 

- The paragraphs in lines 261-313 lack literature references. Idem in line 331: references are given for Denmark, but not to the other mentioned countries.

- The language requires a general revision; two variants of English spelling are used (e.g., British “vapour” and American “vapor”), there are also typos (e.g., line 465 “historick” instead of “historic”) - these mistakes can be easily avoided by using freeware grammar and spell checkers.

Author Response

- Line 68: I suggest substituting “not cultural monuments” with “not classified as cultural monuments” 

Thank you, the recommended language change has been done.

- The paragraphs in lines 261-313 lack literature references. Idem in line 331: references are given for Denmark, but not to the other mentioned countries.

The above-mentioned paragraph is based on information presented in previous authors’ publication [12]. Reference to this publication has been added.

Line 331: reference for Denmark was deleted, as this is only an introductory sentence and individual references are in the following text.

- The language requires a general revision; two variants of English spelling are used (e.g., British “vapour” and American “vapor”), there are also typos (e.g., line 465 “historick” instead of “historic”) - these mistakes can be easily avoided by using freeware grammar and spell checkers.

Thank you, we have tried to unite the spelling and correct the unnecessary mistakes.

Reviewer 2 Report

The authors very honestly supplemented the article based on comments.

However, some aspects remain open to debate. The argument that the case study was not the subject of research can be accepted. However, it would be appropriate to support the claim "Urban tenement houses, which were built in the second half of the 19th and the beginning of the 20th century in the large cities of the Czech Kingdom, form an important part of urban complexes" (line 34), with a schematic map of an urban fragment, to emphasize the relevance of the research. It is commonly known that entire compact urban districts in Prague and other Czech cities are confronted with this problem. In this way, the authors would highlight the relevance of their research. In a way, it would replace the mentioned case study. Moreover, it is an easily accessible data.

But please take this response only as a non-binding recommendation.

Author Response

However, some aspects remain open to debate. The argument that the case study was not the subject of research can be accepted. However, it would be appropriate to support the claim "Urban tenement houses, which were built in the second half of the 19th and the beginning of the 20th century in the large cities of the Czech Kingdom, form an important part of urban complexes" (line 34), with a schematic map of an urban fragment, to emphasize the relevance of the research. It is commonly known that entire compact urban districts in Prague and other Czech cities are confronted with this problem. In this way, the authors would highlight the relevance of their research. In a way, it would replace the mentioned case study. Moreover, it is an easily accessible data.

But please take this response only as a non-binding recommendation.

Thank you, we have added some information supporting the relevance of our research, unfortunately, we do not have the copyright for a suitable historical map of Prague.

Reviewer 3 Report

All elements of the research work are presented somewhere in the text of the manuscript, but still without a clear structure of the scientific work, i.e. without a clear beginning, core and end of the scientific research and important sections related to the way the research was conducted and the discussion of its results in the context of existing research.

Please correct the manuscript so that it has a clear scientific structure and conforms to the Buildings template, which can be found at the following address: https://www.mdpi.com/journal/buildings/instructions

It is necessary to:
1. Adhere to the structure of the summary, which you must clearly define:
(1) Background: Place the question addressed in a broad context and highlight the purpose of the study –  what the current summary contains; (2) Methods: briefly describe the main methods or treatments applied; (3) Results: summa-rize the article’s main findings; (4) Conclusions: indicate the main conclusions or interpretations.

2. The current introduction contains the problem, the focus (the topic) and the importance of the research, but it is missing: The current state of the research field should be carefully reviewed and key publications cited as well as the main aim of the work and highlight the principal conclusions.

3. The presentation of the methodology is still missing. The Materials and Methods should be described with sufficient details to allow others to replicate and build on the published results. Describe the process of how you conducted your own research. Did you obtain the input data from the literature or based on field research? If the building data was obtained based on a field study, indicate the sample size and characteristics (baseline building data). Next, write on what basis you chose the input parameters for the models (e.g., a wall thickness of 45 cm) and why these structures and details were chosen, and then indicate how you analyzed the models. Define what a hygrothermal analysis is, and indicate whether certain software was used for research purposes.

4. Current 2nd (from line 71) and 3rd (from line 178) sections are already part of the results of your research and are solidly written. What is problematic is how the rest of the text in Section 4. Hygrothermal analysis of structures and details is written. This section is about presenting the results of your research, but you include in it a part related to the literature review (the text from line 321 to 339 belongs in the Introduction section, not in the Results), the subsection you call Materials and Model, which should be renamed to better fit the content, and the Discussion section.

5. Write the discussion as a separate section in which you: should discuss the results and how they can be interpreted from the perspective of previous studies and of the working hypotheses. The findings and their implications should be discussed in the broadest context possible. Future research directions may also be highlighted.

6. "Conclusion" section is too general. This section is not mandatory but can be added to the manuscript if the discussion is unusually long or complex. If you are going to write it, state the exact conclusions of your research.

* parts of the text marked in italics are from the Buildings template

Author Response

  1. Adhere to the structure of the summary, which you must clearly define: (1) Background: Place the question addressed in a broad context and highlight the purpose of the study – what the current summary contains; (2) Methods: briefly describe the main methods or treatments applied; (3) Results: summa-rize the article’s main findings; (4) Conclusions: indicate the main conclusions or interpretations.

Necessary updates were made to the article. The authors firmly believe that the presented article is now clear to the reader as well as sufficiently scientific.

  1. The current introduction contains the problem, the focus (the topic) and the importance of the research, but it is missing: The current state of the research field should be carefully reviewed and key publications cited as well as the main aim of the work and highlight the principal conclusions.

Thank you, we have added some information supporting the relevance of our research.

  1. The presentation of the methodology is still missing. The Materials and Methods should be described with sufficient details to allow others to replicate and build on the published results. Describe the process of how you conducted your own research. Did you obtain the input data from the literature or based on field research? If the building data was obtained based on a field study, indicate the sample size and characteristics (baseline building data). Next, write on what basis you chose the input parameters for the models (e.g., a wall thickness of 45 cm) and why these structures and details were chosen, and then indicate how you analyzed the models. Define what a hygrothermal analysis is, and indicate whether certain software was used for research purposes.

The requested information with detail about the hygrothermal analysis was added.

  1. Current 2nd (from line 71) and 3rd (from line 178) sections are already part of the results of your research and are solidly written. What is problematic is how the rest of the text in Section 4. Hygrothermal analysis of structures and details is written. This section is about presenting the results of your research, but you include in it a part related to the literature review (the text from line 321 to 339 belongs in the Introduction section, not in the Results), the subsection you call Materials and Model, which should be renamed to better fit the content, and the Discussion section.
  2. Write the discussion as a separate section in which you: should discuss the results and how they can be interpreted from the perspective of previous studies and of the working hypotheses. The findings and their implications should be discussed in the broadest context possible. Future research directions may also be highlighted
  3. "Conclusion" section is too general. This section is not mandatory but can be added to the manuscript if the discussion is unusually long or complex. If you are going to write it, state the exact conclusions of your research.

Authors deem the Discussion and Conclusion sections to be adequate. Also, other reviewers regarded these sections as solidly written and well formulated.

Reviewer 4 Report

The paper had several improvements with the revision made by the authors. I have a suggestion regarding materials and methods:
- Include more information about the numerical methodology used by the software. The calculation procedure could be checked in the software manual and brief comments included in the text, such as which models are solved (two-dimensional heat conduction?, mass balance?), discretization, and numerical method. Also, more information about the input data, such as boundary conditions.

Author Response

The paper had several improvements with the revision made by the authors. I have a suggestion regarding materials and methods:

- Include more information about the numerical methodology used by the software. The calculation procedure could be checked in the software manual and brief comments included in the text, such as which models are solved (two-dimensional heat conduction?, mass balance?), discretization, and numerical method. Also, more information about the input data, such as boundary conditions.

Thank you. Suggested information with the details about the hygrothermal analysis was added.

Round 3

Reviewer 1 Report

The authors have properly revised the manuscript.

Reviewer 3 Report

The authors have significantly improved the quality of the presentation of the research results. They largely considered all previous comments.